# Genetic-Evolutionary Graph Neural Networks: A Paradigm for Improved Graph Representation Learning

## Abstract

Message-passing graph neural networks have become the dominant framework for learning over graphs. However, empirical studies continually show that message-passing graph neural networks tend to generate over-smoothed representations for nodes after iteratively applying message passing. This over-smoothing problem is a core issue that limits the representational capacity of message-passing graph neural networks. We argue that the fundamental problem with over-smoothing is a lack of diversity in the generated embeddings, and the problem could be reduced by enhancing the embedding diversity in the embedding generation process. To this end, we propose genetic-evolutionary graph neural networks, a new paradigm for graph representation learning inspired by genetic algorithms. We view each layer of a graph neural network as an evolutionary process and develop operations based on crossover and mutation to prevent embeddings from becoming similar to one another, thus enabling the model to generate improved graph representations. The proposed framework has good interpretablility, as it directly draws inspiration from genetic algorithms for preserving population diversity. We experimentally validate the proposed framework on six benchmark datasets on different tasks. The results show that our method significant advances the performance current graph neural networks, resulting in new state-of-the-art results for graph representation learning on these datasets.

## 1 Introduction

Graphs are a general data structure for representing and analyzing complex relationships among entities. Many real-word systems, such as social networks, molecular structures, communication networks, can be modeled using graphs. It is essential to develop intelligent models for uncovering the underlying patterns and interactions within these graph-structured systems. Recent years have seen an enormous body of studies on learning over graphs. The studies include graph foundation models, geometry processing and deep graph embedding. These advances have produced new state-of-the-art or human-level results in various domains, including recommender systems, chemical synthesis, and 2D and 3D vision tasks (Zhang et al., 2024; Xie et al., 2024; Chen et al., 2024; Kim et al., 2023).

Graph neural networks have emerged as a dominant framework for learning from graph-structured data. The development of graph neural network models can motivated from different approaches. The fundamental graph neural network was been derived as a generalization of convolutions to non-Euclidean data (Bruna et al., 2014), as well as by analogy to classic graph isomorphism tests (Hamilton et al., 2017). Regardless of the motivations, the defining feature of the graph neural network framework is that it utilizes a form of message passing wherein messages are exchanged between nodes and updated using neural networks (Hamilton, 2020). During each graph neural network layer, the model aggregates features from a node's local neighbourhood and then updates the node's representation according to the aggregated information.

Message passing is at the heart of current graph neural networks. However, this paradigm of message passing also has major limitations. Theoretically, it is connected to the Weisfeiler-Lehman (WL) isomorphism test as well as to simple graph convolutions. The representational capacity of

message-passing graph neural networks is inherently bounded by the WL isomorphism test. Empirical studies continually find that massage-passing graph neural networks suffer from the problem of over-smoothing. That is, the representations for all nodes can become very similar to one another after too many message passing iterations. These core limitations prevent graph neural networks from more meaningful representations from graphs. In recent years, increasing studies have been devoted to addressing the bottlenecks, such as normalization and regularization techniques (**?**), and combining the global self-attention mechanism (Rampasek et al., 2022), exploring generalized message passing (Barceló et al., 2020). Regardless of these advances, improving the capability of graph neural network models still remains a fundamental challenge in learning from graph-structured data.

To learn meaningful graph representations, it is crucial to generate embeddings for all nodes that depend on both the graph structure and node attributes. However, when the over-smoothing phenomenon occurs, the representations for all nodes begin to look identical to each other. The consequence is that the information from node-specific features becomes lost. To prevent this issue, it is important to perserve the diversity of generated embeddings throughout their layerwisely generation process. *In this paper, we propose genetic-evolutionary graph neural networks, a new paradigm for graph representation learning that integrates the idea from genetic algorithms for maintaining population diversity into the message-passing graph neural network framework.*

Genetic algorithms, inspired by the Charles Darwin's theory of natural evolution, emulate the process of natural selection, wherein the fittest individuals are selected to reproduce and generate the next generation of offspring. Genetic algorithms employ a set of evolution-inspired operations, including mutation, crossover, and selection (Mitchell, 1998). Over multiple generations, biological organisms evolve based on the principle of natural selection, or "survival of the fittest", enabling them to accomplish target tasks. Genetic algorithms have been successfully applied in solving complex optimization and search problems. In machine learning, genetic algorithms have also been used for feature selection (Babatunde et al., 2014) and hyperparameter tuning for models like neural networks and support vector machines (Alibrahim & Ludwig, 2021).

In genetic algorithms, the crossover and mutation operations play a key role in generating diverse individuals for selection, preventing the algorithms from premature convergence (Gupta & Ghafir, 2012). Crossover introduces variety by combining genetic information from different parents, and mutation introduces small random changes in genetic information. In this work, we view the iterative node embedding process as an evolutionary process, in which each layer of message passing produces a new generation of embeddings. We introduce two crossover operations, i.e., cross-generation crossover and sibling crossover, and a mutation operation, and we develop two graph neural network building blocks based on the operations. At each layer of a graph neural network, we first use message passing to update node representations and then apply crossover and mutation to prevent embeddings from becoming similar to one another, thus enabling the model to learn improved graph representations.

Unlike previous methods, such as residual connections (He et al., 2016), SSFG (Zhang et al., 2022) and PairNorm (Zhao & Akoglu, 2020), this work proposes operations by drawing inspiration from genetic algorithms for addressing the over-smoothing problem in graph neural network. Our framework has good interpretablilty as it views the layerwisely node embedding process as analogous to the genetic evolutionary process.. It is a general paradigm that can be integrated into different graph neural network models. We conduct experiments on six benchmark datasets on different graph tasks. We show that the use of our framework significantly improves the performance of the baseline graph neural networks, advancing the state-of-the-art results for graph representation learning on the datasets.

The main contributions of this paper can be summarized as follows. (1) This paper proposes a new framework named genetic-evolutionary graph neural networks for learning from graph-structured data. The core idea behind the proposed framework is to model each layer of a graph neural network as an evolutionary process. We develop three key operations inspired by crossover and mutation from genetic algorithms to enhance the diversity of generated embeddings at each layer. (2) The proposed framework offers good interpretability, as it is directly inspired by biogenetics. It is a general paradigm which can be integrated into current message-passing graph neural networks. Empirical evaluations are conducted on six popular datasets on different graph tasks, and the results demonstrate that the proposed framework significantly improves the performance of the baseline graph neural networks.

## 2 RELATED WORK

### 2.1 GRAPH NEURAL NETWORKS

Most current graph neural networks can be categorized into spectral approaches and spatial approaches (Veličković et al., 2018). The spectral approaches are developed based on spectral graph theory. The key idea of spectral graph neural networks is that convolutions are defined in the spectral domain through an extension of the Fourier transform to graphs. In contrast, spatial graph neural networks define convolutions in spatially localized neighbourhoods. The behaviour of the convolutions is analogous to that of kernels in convolutional neural networks which aggregate features from spatially-defined patches in an image.

Both spectral and spatial graph neural networks are essentially message-passing neural networks that employ a paradigm of message passing wherein embeddings are exchanged between nodes and updated using neural networks (Gilmer et al., 2017). A common issue with message-passing graph neural networks is known as the over-smoothing problem. This issue of over-smoothing was first identified by Li et al. (2018). It can also be viewed as a consequence of the neighbourhood aggregation operation in the message-passing update (Hamilton, 2020). The follow-up studies for limiting over-smoothing include graph normalization and regularization techniques (Zhao & Akoglu, 2020; Chen et al., 2022), combing the global self-attention with local message passing (Rampasek et al., 2022), and improved graph attention approaches (Wu et al., 2024). Additionally, there have been studies on uncovering over-smoothing in basic graph neural network models from theoretical analysis (Oono & Suzuki, 2020). Luan et al. (2024) analyzed homophily by studying intra- and inter-class node distinguishability and showed that graph neural network is capable of generating meaningful representations regardless of homopiily levels.

### 2.2 GENETIC ALGORITHMS

Genetic algorithm methods are inspired by the mechanisms of evolution and natural genetics (Srinivas & Patnaik, 1994). Genetic algorithms were first introduced by Holland (1992) as a heuristic method based on the principle of nature selection. Over the past years, genetic algorithms have emerged as a powerful tool for solving complex optimization and search problems across numerous fields such as scheduling, mathematics and networks (Alhijawi & Awajan, 2023).

In machine learning, genetic algorithms have been applied for optimizing neural networks (Miller et al., 1989) and designing neural network architectures (Jones, 1993). Researchers have also used genetic algorithms for optimizing hyperparameters in neural networks and support vector machines (Alibrahim & Ludwig, 2021; Shanthi & Chethan, 2022). In object detection, hyperparameter evolution which uses a genetic algorithm was applied for optimizing hyperparameters in YOLO models (Redmon, 2016). Sehgal et al. (2019) showed that evolving the weights of a deep nerual network using a genetic algorithm was a competitive approach for training reinforcement learning models.

## 3 METHODOLOGY

### 3.1 GRAPH NEURAL NETWORKS

A graph $\mathcal{G} = (\mathcal{V}, \mathcal{E})$ can be defined through a set of nodes $\mathcal{V}$ and a set of edges $\mathcal{E}$ between pairs of these nodes. Each node $u \in \mathcal{V}$ is associated with a node-level feature $\mathbf{x}_u$. Graph neural networks are a general framework for reorientation learning over the graph $\mathcal{G}$ and $\{\mathbf{x}_u, \forall u \in \mathcal{V}\}$. At its core, the graph neural network framework iteratively updates the representation for every node using a form of message passing. During each message-passing iteration, each node $u \in \mathcal{V}$ aggregates the representations of the nodes in its neighborhood, and the representation for node $u$ is then updated according to the aggregated representation. Following Hamilton (2020), this message-passing framework can be expressed as follows:

$$\mathbf{h}_u^{(k)} = Update^{(k)}\left(\mathbf{h}_u^{(k-1)}, Aggregate^{(k)}(\{\mathbf{h}_v^{(k-1)}, \forall v \in \mathcal{N}(u)\})\right), \qquad (1)$$

where $Update$ and $Aggregate$ are neural networks, and $\mathcal{N}(u)$ is the set nodes in $u$'s neighbourhood. The superscripts are used for distinguishing the embeddings and functions at different iterations. At

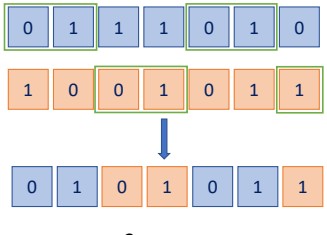 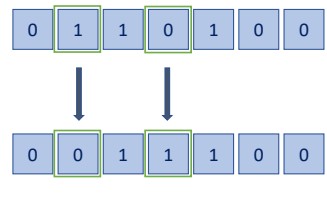

Figure 1: Crossover recombines of the genetic information of parents to produce an offspring. Mutation introduces small random changes in genetic information.

each iteration $k$, the *Aggregate* function takes the set of embeddings of nodes in $\mathcal{N}(u)$ as input and generates an aggregated message $\mathbf{m}_{\mathcal{N}(u)}^{(k)}$. The *Update* function then generates the updated embedding for node $u$ based on the message $\mathbf{m}_{\mathcal{N}(u)}^{(k)}$ and $u$'s previous embedding $\mathbf{h}_u^{(k-1)}$. The embeddings at $k = 0$ are initialized to the node-level features, i.e., $\mathbf{h}_u^{(0)} = \mathbf{x}_u, \forall u \in V$. After $K$ iterations of message passing, every node embedding contains information from its $K$-hop neighborhood.

This message passing formalism is currently the dominant framework for learning over graphs. However, a common issue with message-passing graph neural networks is over-smoothing. The idea of over-smoothing is that the embeddings for all nodes begin to become similar and are relatively uninformative after too many rounds of message passing. This issue of over-smoothing can be viewed as a consequence of the neighborhood aggregation operation. Li et al. (2018) showed that the graph convolution of the basic graph convolutional network model (Kipf & Welling, 2016) can be seen as a special form of Laplacian smoothing that generates the representation for every node using the weighted average of a node's itself and its neighbours' embeddings. But after applying too many rounds of Laplacian smoothing, the representations for all nodes will become indistinguishable from each other. From the graph signal processing perspective, multiplying a signal by high powers of the symmetric normalized adjacency matrix $\mathbf{A_{sym}} = \mathbf{D}^{-\frac{1}{2}}\mathbf{A}\mathbf{D}^{\frac{1}{2}}$, which corresponds to a convolutional filter the lowest eigenvalues, or frequencies, of the symmetric normalized Laplacian $\mathbf{L_{sym}} = 1 - \mathbf{A_{sym}}$. Thus, the simple graph neural network that stacks multiple rounds of graph convolution converges all the node representations to constant values within connected components on the graph, i.e., the "zero-frequency" of the Laplacian (Hamilton, 2020).

### 3.2 GENETIC-REVOLUTIONARY GRAPH NEURAL NETWORKS

#### 3.2.1 MOTIVATION

In the above, we discussed the over-smoothing problem in message-passing graph neural network. We see that the fundamental issue is the loss of diversity of embeddings at each layer throughout the generation process. Thus, we can view the trade-off between model performance and depth of popular graph neural network models from this perspective. Graph neural networks need to model complex relationships and long-term dependencies using more layers to improve the performance. However, using using too many layers will eliminate node-specific features, which leads to significantly reduced model performance.

Graph neural networks generate embeddings for nodes through an iterative message-passing process. At each message-passing iteration, the representation for every node is updated according to the information information aggregated from the node's graph neighbourhood. We can view this iterative process as an genetic evolutionary process, wherein graph nodes are individuals of a population, and the model is to evolve a population of nodes over multiple generations to obtain their expressive representations for graph tasks.

In genetic algorithms, a very homogeneous population, i.e., little population diversity, is considered as the major reason for premature converging to suboptimal solutions (Whitley, 2001). Therefore, it is crucial to preserve the diversity of population during the evolutionary process. Similarly, we

need to maintain the diversity of generated embedding in their generation to prevent the model from converging to a local optimum in optimization.

To preserve the population diversity, genetic algorithms use the operators of crossover and mutation to generate diverse individuals and select those best fit the environment to evolve over successive generations. The crossover operation recombines of the characteristics of each ancestor of an offspring, and the mutation operation randomly changes the genetic information to increase the variability (see Figure 1). In a similar manner, we can generalize the mechanisms to the embedding generation process. By integrating crossover and mutation methods within the message-passing framework, we can prevent generated embeddings from becoming too similar to each other. This ultimately would enhance the model representational capacity.

### 3.2.2 Improving Graph Neural Networks with Genetic Operations

We view each layer of a graph neural network as a genetic evolution process, in which the nodes represent individuals of a population and their embeddings represent chromosomes that store genetic information. During each graph neural network layer, we first use message passing to update the embeddings for all nodes and then use genetic operations to increase the diversity of generated embeddings. We propose three operations inspired by genetic algorhtms: (1) cross-generation crossover, (2) sibling crossover, and (3) mutation.

Genetically, crossover is a process in which the genetic information of two parents is recombined to produce new offspring, resulting in the exchange of genetic material between parental chromosomes. This mechanism forms the basis for driving biological variation, shaping differences in traits within species and introducing novel traits previously unseen in a population. It basically helps promote the evolutionary process by enabling novel gene combinations to emerge and spread across generations. Fundamentally, this process creates diversity at the level of genes that reflects difference in chromosomes of different individuals.

**Cross-generation crossover**. Similar to crossover in genetics, the cross-generation operation in our framework recombines the embedding for a node generated by message-passing and the node's previous layer embedding. For $\overline{\mathbf{h}}_u^{(k)} = (\overline{\mathbf{h}}_{u,1}^{(k)}, ..., \overline{\mathbf{h}}_{u,d}^{(k)})$ and $\mathbf{h}_u^{(k-1)} = (\mathbf{h}_{u,1}^{(k-1)}, ..., \mathbf{h}_{u,d}^{(k-1)})$ which represent the embedding for node $u$ generated by message passing and $u$'s previous layer embedding, cross-generation crossover can be expressed as follows:

$$\mathbf{h}_u^{(k)} = Crossover(\overline{\mathbf{h}}_u^{(k)}, \mathbf{h}_u^{(k-1)})$$
$$\text{where } \mathbf{h}_{u,i}^{(k)} = \begin{cases} \mathbf{h}_{u,i}^{(k)} & \text{if } \lambda_i < p \\ \overline{\mathbf{h}}_{u,i}^{(k)} & \text{else} \end{cases}, \tag{2}$$

and $\lambda_i \sim U(0,1)$ and $p$ is a probability indicating information from the previous layer embedding. At each dimension, the feature is randomly selected from the embedding generated using message passing or from the embeding inputted to this layer. Because each round of message passing generates a smoothed version of the input, recombining information from a node's previous layer embedding reduces the smoothness of the generated embeddings. This operation is a parameter-free method and can be integrated into current graph nerual networks.

**Sibling crossover** is an operation that randomly selects information from siblings. In our impelmentation, we generate multi-head outputs using message passing as siblings and update the embedding for a node by randomly selecting information from the multi-head outputs.

$$\mathbf{h}_u^{(k)} = Crossover(\overline{\mathbf{h}}_u^{(k,head_1)}, ..., \overline{\mathbf{h}}_u^{(k,head_z)})$$
$$\text{where } \mathbf{h}_{u,i}^{(k)} = \overline{\mathbf{h}}_{u,i}^{(k,h_{ij})}, \tag{3}$$

$h_{ij} \sim Categorical(\frac{1}{z}, ..., \frac{1}{z})$, and $z$ is the number of heads. Each $\overline{\mathbf{h}}_u^{(k,head_h)}$ in the multi-head outputs represents a sibling generated using the same input. This operation also increases individual diversity by randomly combining information from different siblings.

**Mutation** is the process in which some genes of individuals are randomly changed. In our framework, the feature at each dimension is randomly replaced by a value sampled from a Gaus-

---

**Algorithm 1** Pseudocode for cross-generation crossover in a PyTorch-like style.

---

```python
# h, h_in: representaton generated by message passing and the previous layer embedding
# f_prob: probabilty of recombining information from parent
# self.dist: a Bernoulli distribution defined by torch.distributions.Bernoulli(torch.tensor(
    self.f_prob)):

def forward(self, h, h_in):

    if self.training == True:
        crossover_mask = self.dist.sample(h.shape) # generate crossover mask

        # crossover from h and h_in
        h = h_in * crossover_mask + h * (1 - crossover_mask)
    else:
        h = h_in * self.f_prob + h * (1 - self.f_prob)

    return h
```

---

**Algorithm 2** Pseudocode for mutation in a PyTorch-like style.

---

```python
# self.running_mean: the mean of h over the training set
# self.running_var: the variance of h over the training set
# self.mutation_prob: probility of mutation

def forward(self, h):
    if self.training == True:
        mean = h.mean([0])
        var = h.var([0])
        n = h.numel() / h.size(1)

        with torch.no_grad():
            # momentum update of running_mean and running_var
            self.running_mean = self.momentum * mean + (1 - self.momentum) * self.running_mean
            self.running_var = self.momentum * var * n / (n - 1) + (1 - self.momentum) * self.
                running_var

    # generate mutatioin noise
    gaussian_noise = torch.randn(h.shape)

    if self.training == True:
        mutation_mask = Bernoulli.sample(h.shape) # generate mutation mask
        h = (gaussian_noise * self.running_var + self.running_mean) * mutation_mask + h * (1 -
            mutation_mask)
    else:
        h = self.running_mean * self.mutation_prob + h * (1 - self.mutation_prob)

    return h
```

---

sian distribution, wherein the statistics are calculated using batches. For a batch of $m$ vectors $\mathcal{B} = \{\mathbf{h}_u^1, \mathbf{h}_u^2, ..., \mathbf{h}_u^m\}$, we calculate the mean $\boldsymbol{\mu}$ and variance $\boldsymbol{\delta}$ of the feature over the training set as follows.

$$\boldsymbol{\mu} \leftarrow \mathbb{E}_{\mathcal{B}}(\mu_{\mathcal{B}})$$
$$\boldsymbol{\delta} \leftarrow \frac{m}{m-1}\mathbb{E}_{\mathcal{B}}(\delta_{\mathcal{B}}^2) \qquad (4)$$

where $\mu_{\mathcal{B}}$ and $\delta_{\mathcal{B}}^2$ are the mean and variance of the batch $\mathcal{B}$. Here we use the unbiased variance estimate. Then we randomly sample a vector $\boldsymbol{\gamma}$ from a multivariate Gaussian distribution $N(\mathbf{0}, \mathbf{I})$ and update the feature as follows:

$$\tilde{\mathbf{h}}_u^i = (\boldsymbol{\gamma}\boldsymbol{\delta} + \boldsymbol{\mu})\mathbf{mask} + \mathbf{h}_u^i(1 - \mathbf{mask}) \qquad (5)$$

where the $\mathbf{mask} \sim Bernoulli(mutation\_rate)$. The mutation operation is also a parameter-free method. It basically introduces randomness to features as a regularization method, enabling the model to explore new space for optimization.

### 3.3 MODEL ARCHITECTURE

Algorithm 1 and Algorithm 2 show our Pytorch-style pseudo-code for the cross-generation crossover operation and mutation operation respectively. The code for sibling crossover can be easily adapted from Algorithm 1. We design two building blocks based on the cross-generation crossover operaton and sibling crossover operation (see Figure 2). The first building block applies the cross-generation

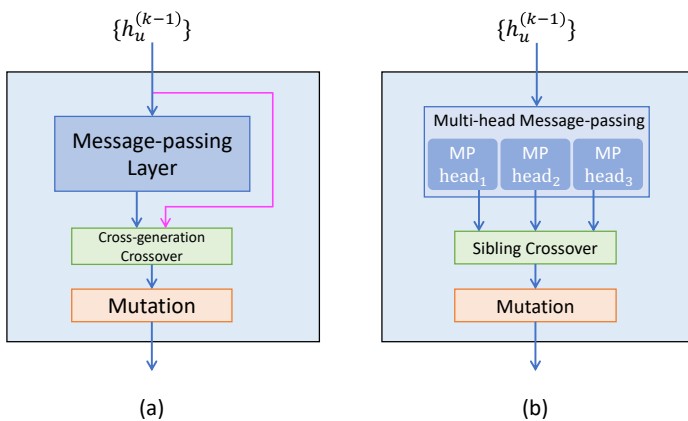

Figure 2: Building block architectures: Block (a) applies cross-generation to a node's embedding generated using message passing and the node's previous layer embedding, and Block (b) applies sibling crossover to a set of outputs generated using multi-head message passing.

Table 1: Classification accuracy (%) on MNIST and CIFAR10 on the superpixel graph classification task. The cross-generation crossover and mutation operations are applied to the base GPS model.

| Model | MNIST | CIFAR10 |
|---|---|---|
| GCN (Kipf & Welling, 2016) | 90.705±0.218 | 55.710±0.381 |
| MoNet (Monti et al., 2017) | 90.805±0.032 | 54.655±0.518 |
| GraphSAGE (Hamilton et al., 2017) | 97.312±0.097 | 65.767±0.308 |
| GIN (Xu et al., 2019) | 96.485±0.252 | 55.255±1.527 |
| GCNII (Chen et al., 2020) | 90.667±0.143 | 56.081±0.198 |
| PNA (Corso et al., 2020) | 97.94±0.12 | 70.35±0.63 |
| DGN (Beaini et al., 2021) | – | 72.838±0.417 |
| CRaWl (Toenshoff et al., 2021) | 97.944±0.050 | 69.013±0.259 |
| GIN-AK+ (Zhao et al., 2021) | – | 72.19±0.13 |
| 3WLGNN (Maron et al., 2019) | 95.075±0.961 | 59.175±1.593 |
| EGT (Hussain et al., 2022) | 98.173±0.087 | 68.702±0.409 |
| GatedGCN + SSFG (Zhang et al., 2022) | 97.985±0.032 | 71.938±0.190 |
| EdgeGCN (Zhang et al., 2023) | 98.432±0.059 | 76.127±0.402 |
| Exphormer (Shirzad et al., 2023) | 98.550±0.039 | 74.754±0.194 |
| TIGT (Choi et al., 2024) | 98.230±0.133 | 73.955±0.360 |
| RandAlign + GatedGCN (Zhang & Xu, 2024) | 98.512±0.033 | 76.395±0.186 |
| GCN (Rampasek et al., 2022) | 90.705±0.218 | 55.710±0.381 |
| **Ours + GCN** | **95.926±0.031** | **59.157±0.130** |
| GPS (Rampasek et al., 2022) | 98.051±0.126 | 72.298±0.356 |
| Finetuned GPS | 98.186±0.107 | 75.680±0.188 |
| **Ours + Finetuned GPS** | **98.685±0.029** | **80.636±0.195** |

crossover after message passing, followed by the mutation operation. Note that this building block is compatible with different graph neural network models and it does not introduce additional trainable parameters. The other building block applies sibling crossover to a set of multi-head outputs, followed by the mutation operation. This method requires the model to generate multiple siblings using a multi-head message passing.

The embedding generation process takes the graph $\mathcal{G} = (\mathcal{V}, \mathcal{E})$ and features for all nodes $\mathbf{x}_u, \forall u \in \mathcal{V}$, as input. This is followed by $K$ building blocks that generate hidden embeddings. Finally, a readout function is applied to the output of the last block to generate the graph representation. For node-level tasks, the embeddings generated by the last block are directly used.

Table 2: Results on PascalVOC-SP and COCO-SP on the node classification task. The cross-generation crossover and mutation operations are applied to the base GPS model.

| Model | PascalVOC-SP (F1) | COCO-SP (F1) |
|---|---|---|
| GCN Kipf & Welling (2016) | 0.1268±0.0060 | 0.0841±0.0010 |
| GINE Hu et al. (2019) | 0.1265±0.0076 | 0.1339±0.0044 |
| GCNII Chen et al. (2020) | 0.1698±0.0080 | 0.1404±0.0011 |
| GatedGCN Bresson & Laurent (2017) | 0.2873±0.0219 | 0.2641±0.0045 |
| GatedGCN + RWSE (Rampasek et al., 2022) | 0.2860±0.0085 | 0.2574±0.0034 |
| Transformer + LapPE Dwivedi et al. (2022) | 0.2694±0.0098 | 0.2618±0.0031 |
| SAN + LapPE Dwivedi et al. (2022) | 0.3230±0.0039 | 0.2592±0.0158 |
| SAN + RWSE Dwivedi et al. (2022) | 0.3216±0.0027 | 0.2434±0.0156 |
| Exphormer Shirzad et al. (2023) | 0.3975±0.0037 | 0.3455±0.0009 |
| RandAlign + GPS (Zhang & Xu, 2024) | 0.4242±0.0011 | 0.3567±0.0026 |
| Fine-tuned GCN (Tönshoff et al., 2023) | 0.2078±0.0031 | – |
| **Ours + Finetuned GCN** | **0.2241±0.0020** | – |
| GPS (Rampasek et al., 2022) | 0.3748±0.0109 | 0.3412±0.0044 |
| Fine-tuned GPS (Tönshoff et al., 2023) | 0.4440±0.0065 | 0.3884±0.0055 |
| **Ours + Finetuned GPS** | **0.4832±0.0031** | **0.4002±0.0019** |

Table 3: Results on Pepti-func and Pepti-struct. The sibling crossover and mutation operations are applied to the base GCN model.

| Model | Peptides-func (AP ↑) | Peptides-struct (MAE ↓) |
|---|---|---|
| GCN | 0.5930±0.0023 | 0.3496±0.0013 |
| GINE | 0.5498±0.0079 | 0.3547±0.0045 |
| GCNII (Chen et al., 2020) | 0.5543±0.0078 | – |
| GatedGCN | 0.5864±0.0077 | 0.3420±0.0013 |
| Gated + RWSE | 0.6069±0.0035 | 0.3357±0.0006 |
| Transformer+LapPE | 0.6326±0.0126 | 0.2529±0.0016 |
| SAN+LapPE | 0.6384±0.0121 | 0.2683±0.0043 |
| SAN+RWSE | 0.6439±0.0075 | 0.2545±0.0012 |
| Exphormer (Shirzad et al., 2023) | 0.6527±0.0043 | 0.2481±0.0007 |
| GPS (Rampasek et al., 2022) | 0.6535±0.0041 | 0.2500±0.0005 |
| Finetuned GPS (Tönshoff et al., 2023) | 0.6534±0.0091 | 0.2509±0.0014 |
| Finetuned GCN (Tönshoff et al., 2023) | 0.6860±0.0050 | 0.2460±0.0007 |
| **Ours + Finetuned GCN** | **0.7021±0.0034** | **0.2426±0.0014** |

## 4 EMPIRICAL EVALUATION

### 4.1 DATASETS AND SETUP

The experiments are conducted on six benchmark datasets, i.e., MNIST, CIFAR10, PascalVOC-SP, COCO-SP, Peptides-func and Peptides-struct (Dwivedi et al., 2020; 2022) on three graph tasks, graph classification, node classification, and graph regression. We closely follow the setup as Dwivedi et al. (2020; 2022) for training and evaluating the models. The details of the datasets and evaluation metrics are provided in the appendix section.

### 4.2 RESULTS

**CIFAR10 and MNIST**. Table 1 reports the results on the two datasets on the superpixel classification task. We use the GPS (Rampasek et al., 2022) as the base model. The GPS model is a hybrid of local aggregation and global aggregation architecture. It uses GatedGCN for local aggregation and

Table 4: Ablation study: Importance of crossover and mutation on the model performance on CIFAR10 and PascalVOC-SP.

| Base Model | Crossover | Mutation | CIFAR10 | PascalVOC-SP |
|---|---|---|---|---|
| Finetuned GPS (Tönshoff et al., 2023) | × | × | 75.680±0.188 | 0.4440±0.0065 |
| | ✓ | × | 79.434±0.228 | 0.4952±0.0098 |
| | × | ✓ | 77.029±0.203 | 0.4554±0.0077 |
| | ✓ | ✓ | 80.636±0.195 | 0.4832±0.0031 |

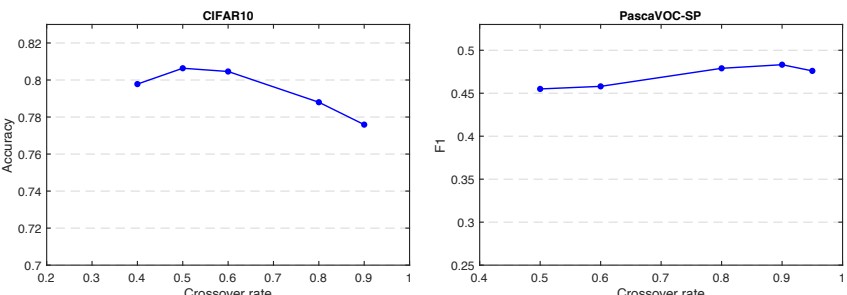

Figure 3: Impact of the crossover rate $p$ on the model performance on CIFAR10 and PascalVOC-SP.

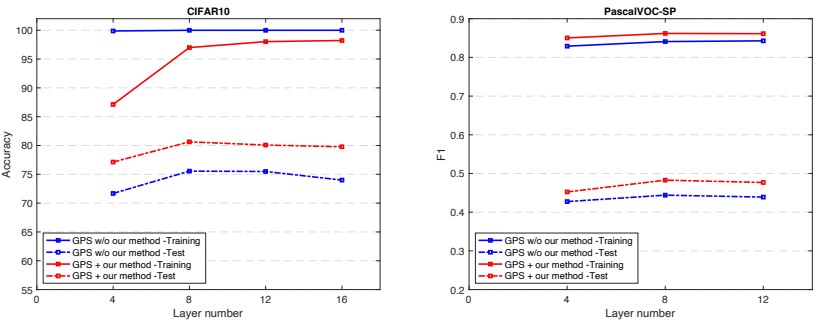

Figure 4: Results of our method on the base Finetuned GPS model with different layers on CIFAR10 and PascalVOC-SP.

uses Transformer for global aggregation. We apply cross-generation and mutation (i.e., block (a) in Figure 2) to the base GatedGCN model. The crossover rate is set to 0.5 and mutation rate is set to 0.1. We see from Table 1 that our method improves the performance of the base model by a large margin, with a relative improvement of 0.648% and 11.53% on MNIST and CIFAR10 respectively. It simultaneously outperforms both Exphormer (Shirzad et al., 2023) and RandAlign (Zhang & Xu, 2024), which previously achieved the best performance on MNIST and CIFAR10 respectively.

**PascalVOC-SP and COCO-SP**. The two datasets are long-range prediction datasets compared to MNIST and CIFAR10. The task is to predict if a node corresponds to a region of an image which belongs to a particular class. We use Finetuned GPS (Tönshoff et al., 2023) as the base model. The Finetuned GPS is also a hybrid of GatedGCN and Transformer architecture. We apply cross-generation and mutation to the base GatedGCN model. The crossover rate is set to 0.9 and mutation rate is set to 0.05. The results are reported in Table 2. Previously, Finetuned GPS achieved the best performance among the baseline models on the two datasets. As compared to Finetuned GPS, the use of our method results in a relative improvement of 8.83% and 3.04% respectively without using additional model parameters. Once again, our framework achieves new state-of-the-art performance on the two datasets.

**Peptides-func and Peptides-struct**. We use Finetuned GCN (Tönshoff et al., 2023) as the base model on the two datasets. We use sibling crossover and mutation to the base model. The number

Table 5: Comparison of our method with the basic GCN, wherein residual connections and batch normalizations (BN) are not used.

| Model | MNIST | CIFAR10 |
|---|---|---|
| GCN (w/o residual connections and BN) | 87.590±0.336 | 48.810±1.045 |
| GCN (with residual connections and BN) | 90.705±0.218 | 55.710±0.381 |
| GCN (with residual connections and BN) + Ours | 95.926±0.031 | 59.157±0.130 |

of siblings is set to 2 and mutation rate is set to 0.1. The results are reported in Table 3. Finetuned GCN is a strong baseline model in previous work. We see from Table 3 that the use of framework further improve the model performance.

**Ablation Study**. We conduct an ablation study on CIFAR10 and PascalVOC-SP to analyse the importance of crossover and mutation on the model performance. Table 5 shows the ablation study results. It can be seen from Table 5 that the crossover operation plays a major role in improving the model performance. The mutation operation helps further improve the model performance as a regularization method.

We further analyzed the impact of the crossover rate $p$ on model performance on CIFAR10 and PascalVOC-SP. Figure 3 shows the experimental results. We see that the best performance is achieved when $p$ is set to different values on the two datasets. When $p$ is set to 0, it is equivalent to not using crossover. A recommended strategy for tuning $p$ is starting from 0.9 or 0.95 and then gradually decreasing it to find the optimal value.

We conducted experiments to analysis the performance of our method on the base Finetuned GPS model with different layers on CIFAR10 and PascalVOC-SP. The results are shown in Figure 4. We also analyzed the the performance of our method on the base Finetuned GPS model with different layers on CIFAR10, and the results are reported in Figure 5 in the appendix section. It can be seen from Figure 4 and Figure 5 that the use of our method improves the model generalization performance on the base models.

We further compared our method with the basic GCN in which residual connections and batch normalizations are not used on MNIST and CIFAR10. The results are shown in Table 5. We see that the model performance drops without using these techniques and that the joint use of our method with residual connections and batch normalizations yields the best task perforamnce.

## 5 CONCLUSIONS

This paper presents a new framework called genetic-evolutionary graph neural networks for graph representation learning. The key idea of our approach is to view each layer of a graph neural network as a genetic evolutionary process and use biogenetics-inspired operations to prevent the oversmoothing problem in graph neural networks. We developed three operations, i.e., cross-generation crossover, sibling crossover and mutation, inspired by genetic algorithms and presented two building blocks based on the the operations for graph representation learning. An important advantage of the proposed framework lies in its interpretability, as it frames layerwisely graph representation learning as an evolutionary process. The experimental evaluations were conducted on six popular datasets on different graph tasks. The results showed that the use of our framework significantly improves the performance of the base graph neural networks, achieving new state-of-the-art performance for graph representation learning on these datasets. We also presented ablations of our framework, showing the importance of each operation on the overall model performance.

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

# A  APPENDIX

**Datasets**. The experiments were conducted on the following six benchmark datasets.

- **MNIST and CIFAR10** are two datasets for superpixel graph classification (Dwivedi et al., 2020). The superpixels are converted from original images in MNIST (LeCun et al., 1998) and CIFAR10 (Krizhevsky et al., 2009) using the SLIC algorithm (Achanta et al., 2012).
- **PascalVOC-SP and COCO-SP** are two datasets of superpiexels (Dwivedi et al., 2022), which are converted from images in original PascalVOC and COCO datasets. The task on the two datasets is to predict if a node corresponds to a region of an image which belongs to a particular class.
- **Peptides-func and Peptides-Struct** (Dwivedi et al., 2022) are two datasets of peptides molecular graphs. The nodes in the graphs represent heavy (non-hydrogen) atoms of the peptides, and the edges represent the bonds between these atoms. The graphs are categorized into 10 classes based on the peptide functions, e.g., antibacterial, antiviral, cell-cell communication. The two datasets are used for evaluating the model's performance for multi-label graph classification and multi-label graph regression.

The statistics of the benchmark datasets used in the experiments are shown in below Table 6.

Table 6: Statistics of the six benchmark datasets used in the experiments.

| Dataset | Graphs | Nodes | Avg. nodes/graph | #Training | #Validation | #Test | #Categories |
|---------|--------|-------|------------------|-----------|-------------|-------|-------------|
| MNIST | 70K | – | 40-75 | 55,000 | 5000 | 10,000 | 10 |
| CIFAR10 | 60K | – | 85-150 | 45,000 | 5000 | 10,000 | 10 |
| PascalVOC-SP | 11,355 | 5,443,545 | 479.40 | 8,489 | 1,428 | 1,429 | 20 |
| COCO-SP | 123,286 | 58,793,216 | 476.88 | 113,286 | 5,000 | 5,000 | 81 |
| Peptides-func | 15,535 | 2,344,859 | 150.94 | 70% | 15% | 15% | 10 |
| Peptides-struct | 15,535 | 2,344,859 | 150.94 | 70% | 15% | 15 | – |

**Evaluation Metrics**. Following Dwivedi et al. (2020) and Rampasek et al. (2022), the following metrics are used evaluation on different tasks. The performance on MNIST and CIFAR10 on graph classification is evaluated using the classification accuracy. The performance on PascalVOC-SP and COCO-SP on node classification is evaluated using the macro weighted F1 score. The performance on Peptides-func on multi-label graph classification is evaluated using average precision (AP) across the categories. The performance on Peptides-struct on multi-label graph regression is evaluated using mean absolute error (MAE).

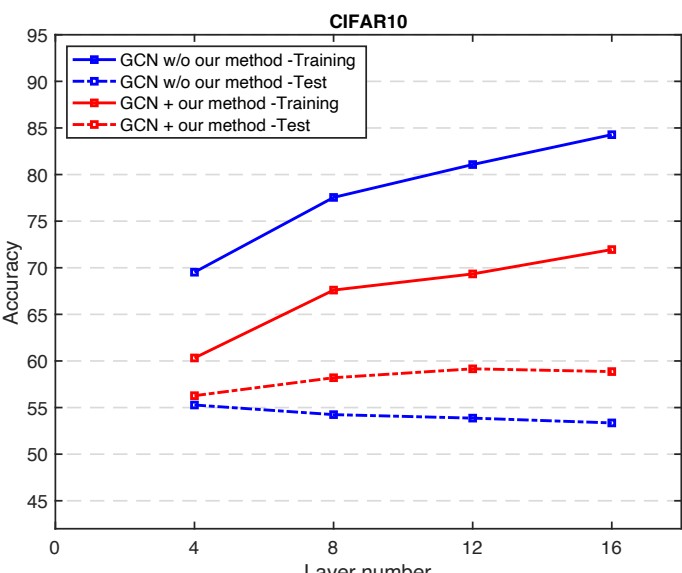

Figure 5: Results of our method on the base Finetuned GCN model with different layers on CIFAR10.

```python
 7 class Crossover(nn.Module):
 8
 9     def __init__(self, f_prob=0):
10         super(Crossover, self).__init__()
11         self.f_prob = f_prob  #prob of sampling 1. for Bern
12         self.mm = torch.distributions.bernoulli.Bernoulli(torch.tensor([self.f_prob],
    device='cuda'))
13
14     def forward(self, h, h_in):
15         if self.training:
16             crossover_mask = self.mm.sample(h.shape).squeeze(-1)
17             h = h_in * crossover_mask + h * (1 - crossover_mask)
18         else:
19             h = h_in * self.f_prob + h * (1- self.f_prob)
20
21         return h
```

Figure 6: Implementation of the crossover operation in Pytorch.

```python
25 class Mutation(nn.BatchNorm1d):
26     def __init__(self, num_features, mutate_prob, eps=1e-5, momentum=0.1,
27                  affine=True, track_running_stats=True):
28         super(Mutation, self).__init__(
29             num_features, eps, momentum, affine, track_running_stats)
30         self.mutate_prob = mutate_prob
31
32     def forward(self, input):
33         self._check_input_dim(input)
34         exponential_average_factor = 0.0
35
36         if self.training and self.track_running_stats:
37             if self.num_batches_tracked is not None:
38                 self.num_batches_tracked += 1
39                 if self.momentum is None:  # use cumulative moving average
40                     exponential_average_factor = 1.0 / float(self.num_batches_tracked
    )
41                 else:  # use exponential moving average
42                     exponential_average_factor = self.momentum
43
44         # calculate running estimates
45         if self.training:
46             mean = input.mean([0])
47             # use biased var in train
48             var = input.var([0], unbiased=False)
49             n = input.numel() / input.size(1)
50             with torch.no_grad():
51                 self.running_mean = exponential_average_factor * mean\
52                     + (1 - exponential_average_factor) * self.running_mean
53                 # update running_var with unbiased var
54                 self.running_var = exponential_average_factor * var * n / (n - 1)\
55                     + (1 - exponential_average_factor) * self.running_var
56         else:
57             mean = self.running_mean
58             var = self.running_var
59         mean = self.running_mean
60         var = self.running_var
61
62         gaussion_noise = torch.randn(input.shape).type_as(input)
63
64         prob_mutate = self.mutate_prob
65         if self.training:
66             mm = torch.distributions.bernoulli.Bernoulli(torch.tensor([prob_mutate],
    device='cuda'))
67             mutate_mask = mm.sample(input.shape).squeeze(-1)
68             input = (gaussion_noise * var + mean)*mutate_mask + input * (1 - mutate_
    mask)
69         else:
70             input = mean * prob_mutate + input * (1 - prob_mutate)
71
72         return input
```

Figure 7: Implementation of the mutation operation in Pytorch.

**Algorithm 3** Pseudo for the cross-generation crossover operation.

**Input:** Crossover probability $p$, **h**, **h_in** // embeddings generated by the current layer and the previous layer

**Output:** **h_crossover** // Crossover of **h** and **h_in**

1: **if** model.training == True **then**
2:    **crossover_mask** = **Bernoulli.sample**(prob=$p$) // each element in **crossover_mask** is sampled from the Bernoulli distribution with probability $p$
3:    **h_crossover** = **h_in** $*$ **crossover_mask** + **h** $* (1 -$ **crossover_mask**$)$
4: **else**
5:    **h_crossover** = **h_in** $* p +$ **h** $* (1 - p)$
6: **end if**

**Algorithm 4** Pseudo for the mutation operation.

**Input:** Node embedding **h**, mutatiion probability $r$

**Output:** **h_mutation** // Mutation output of **h**

1: **running_mean**, **running_var** = Update(**h**) // update running mean and var
2: **gaussian_noise** = Gaussian.Sample() //the reparameterization trick
3: **if** model.training == True **then**
4:    **mutation_mask** = **Bernoulli.sample**(prob=$r$) // each element in **mutation_mask** is sampled from the Bernoulli distribution with probability $p$
5:    **h_mutation** = (**gaussian_noise** $*$ **running_var** + **running_mean**) $*$ **mutation_mask** + **h** $* (1 -$ **mutation_mask**$)$
6: **else**
7:    **h_mutation** = **running_mean** $* r +$ **h** $* (1 - r)$
8: **end if**

