# OpenReview forum: "Genetic-evolutionary Graph Nerual Networks: A Paradigm for Improved Graph Representation Learning"
_ICLR.cc/2025/Conference — Submitted to ICLR 2025_

### Official Review · Reviewer_uj5y · 2024-10-31

**Soundness:** 3
**Presentation:** 4
**Contribution:** 3
**Rating:** 6
**Confidence:** 4

**Summary:**

The paper introduces a novel framework for graph neural networks (GNNs) called Genetic-Evolutionary Graph Neural Networks (GE-GNNs), aimed at enhancing graph representation learning by addressing the issue of over-smoothing in GNNs. Over-smoothing is a phenomenon where node representations become too similar, limiting the model’s effectiveness in distinguishing between different nodes. The authors propose to mitigate this by using genetic algorithms, specifically through evolutionary mechanisms such as crossover and mutation operations.

**Strengths:**

1. The paper clearly identifies the key issue of GNNs—over-smoothing—and proposes using genetic algorithms to address this problem.

2. The framework primarily relies on evolutionary operations, which is a compelling idea.

3. The evaluation appears very solid, covering different tasks and datasets.

**Weaknesses:**

1. The evaluation could be more balanced. GPS is a strong GNN model, and the authors should demonstrate that the proposed framework performs well with other GNN architectures, such as applying genetic algorithms with GCN.

2. The authors could consider adding more detailed descriptions of how the evolutionary operators function within the proposed framework. From my understanding, this framework operates similarly to other GNNs optimized by backpropagation, while evolutionary algorithms (EAs) traditionally do not rely on backpropagation. In this context, the evolutionary operators seem to act more like activation functions or non-parametric layers, primarily generating diverse outputs that are subsequently optimized through backpropagation. Clarifying this distinction would help readers better understand the role and novelty of these operators within the framework.

3. If I missed any details, please correct me. There are two types of crossover operators—why didn’t the authors consistently apply both across all datasets?

4. The ablation study is not convincing.

5. How many repeated experiments did you run, considering that crossover and mutation introduce randomness? Are all results statistically significantly different?

**Questions:**

see above

---

> ### Author Response · Authors · 2024-11-28
> **Response to Reviewer uj5y**
>
> **1. The evaluation could be more balanced. GPS is a strong GNN model, and the authors should demonstrate that the proposed framework performs well with other GNN architectures, such as applying genetic algorithms with GCN.**
>
> *Response:* We have experimented of  our method on GCN on MNIST, CIFAR10 and PascalVOC-SP.
> The results have been added in Table 1 and Table 2 in the revised version.
> We have also experimented of our method on GCN with different layers CIFAR10.
> The results are reported in Figure 5 in the appendix section.
> It can be seen from Figure 5 that our method effectively improves the generalization performance compared to the base model.
>
>
> **2. The authors could consider adding more detailed descriptions of how the evolutionary operators function within the proposed framework. From my understanding, this framework operates similarly to other GNNs optimized by backpropagation, while evolutionary algorithms (EAs) traditionally do not rely on backpropagation. In this context, the evolutionary operators seem to act more like activation functions or non-parametric layers, primarily generating diverse outputs that are subsequently optimized through backpropagation. Clarifying this distinction would help readers better understand the role and novelty of these operators within the framework.**
>
> *Response:* Thank you.
> The operations of crossover and mutation, which are inspired by genetic algorithms, apply on node embeddings.
> The crossover operation can be implemented using Algorithm 1.
> For the mutation operation, it can be implemented using the reparameterization technique.
> Therefore, the two kinds operations can be integrated into current graph neural networks, and the model can be optimized using backkpropagation.
>
> We have added the implementation of the two operation in Pytorch in Figure 6 and Figure 7 in the appendix section.
> Hopefully, the codes could be helpful for reproducing the results and for our framework to be better understood.
>
>
> **3. If I missed any details, please correct me. There are two types of crossover operators—why didn’t the authors consistently apply both across all datasets?**
>
> *Response:* Thank you. The cross-generation crossover operation and mutation operation are parameter-free methods and do not introduce additional trainable model parameters.
> The results in Table 1 and Table 2 are obtained by applying the two operations on base GPS mode.
> We think it is a good way to the performance improvements that are made by using our method.
> For results in Table 3, we found that the use of cross-generation could only yiled marginally improved performance.
> Therefore we chose to use sibling crossover and mutation, the use of sibling crossover increases model parameters because multi-head output need to be generated.
> We appreciate your suggestion, and we understand that the joint use of the two crossover operations could potentially improve the overall performance. Due to the limitation of time, we have not obtained the results.
> However, we will be continuing working on this.
>
>
> **4. The ablation study is not convincing.**
>
> *Response:* We have further conducted experiments to analysis the performance of our method on
> the base Finetuned GPS model with different layers on CIFAR10 and PascalVOC-SP. The results
> have been added in Figure 4 in the revised version. We have also analyzed the the performance of
> our method on the base Finetuned GPS model with different layers on CIFAR10. The results have
> been added in Figure 5 in the appendix section. It can be seen from Figure 4 and Figure 5 that the
> use of our method improves the model generalization performance on the base models
>
> **5. How many repeated experiments did you run, considering that crossover and mutation introduce randomness? Are all results statistically significantly different?**
>
> *Response:* We followed the experimental setup of GPS (Rampasek et al., 2022), 10 runs of experiments are conducted for   and the mean and standard deviation are reported as the model performance.
> Although our method introduces randomness, we can see from the experimental results that our method yields a lower standard deviation value compared to the base models.
> This suggests that our method improves the stability of optimization.

---

> > ### Comment · Reviewer_uj5y · 2024-11-28
> > **Responses**
> >
> > Thank you for your experiments. I find this research idea both promising and interesting. However, I still have a few concerns:
> >
> > (1) The proposed approach employs evolutionary operators as a perturbation-adding process. While the experimental results are generally strong, the approach is designed to address the oversmoothing problem in GNNs. Therefore, it would be beneficial to test the method on a broader range of GNN architectures, rather than restricting the experiments to GCN and GPS. In other words, the focus should be on demonstrating improvements across various GNNs, rather than simply comparing GCN+GA and GPS+GA with other methods. We need to see any GNNs + GA have improvements.
> >
> > (2) The observed performance improvements may not directly result from mitigating oversmoothing. Instead, it is possible that the evolutionary operators are functioning similarly to dropout layers, thereby reducing overfitting. Have you conducted any analysis of oversmoothing using metrics such as Mean Absolute Error (MAE) or Dirichlet Energy?

---

> > > ### Author Response · Authors · 2024-12-02
> > > **Response to Reviewer uj5y**
> > >
> > > **(1) The proposed approach employs evolutionary operators as a perturbation-adding process. While the experimental results are generally strong, the approach is designed to address the oversmoothing problem in GNNs. Therefore, it would be beneficial to test the method on a broader range of GNN architectures, rather than restricting the experiments to GCN and GPS. In other words, the focus should be on demonstrating improvements across various GNNs, rather than simply comparing GCN+GA and GPS+GA with other methods. We need to see any GNNs + GA have improvements.**
> > >
> > > *Response:* Thank you very much for the comments. We have further evaluated our method on GAT
> > > and GatedGCN on the superpixel graph classification datasets of MNIST and CIFAR10. The results
> > > are reported below. It can be seen that our method consistently improve the performance of the
> > > baseline models. Notably, our method on GatedGCN improves the performance from 67.312% to
> > > 78.242%, which is a 16.24% relative improvement.
> > > | Model | MNIST | CIFAR10 |
> > > |-----------|-----------|-----------|
> > > | GCN| 90.705±0.218| 55.710±0.381 |
> > > | **GCN + Ours** | **95.926±0.031**| **59.157±0.130**|
> > > |GAT |95.535±0.205| 64.223±0.455|
> > > |**GAT + Ours** | **97.643±0.108** | **69.247±0.248**|
> > > |GatedGCN| 97.340±0.143 |67.312±0.311|
> > > |**GatedGCN + Ours**| **98.526±0.031**| **78.242±0.239**|
> > > |Fine-tuned GPS| 98.186±0.107 | 75.680±0.188|
> > > |**Fine-tuned GPS + Ours** |**98.685±0.029** | **80.636±0.195**|
> > >
> > > **(2) The observed performance improvements may not directly result from mitigating oversmoothing. Instead, it is possible that the evolutionary operators are functioning similarly to dropout layers, thereby reducing overfitting. Have you conducted any analysis of oversmoothing using metrics such as Mean Absolute Error (MAE) or Dirichlet Energy?**
> > >
> > > *Response:*  Thanks a lot for the comment. We evaluated the method of Dropout on MNIST and
> > > CIFAR10 and found that the use of Dropout achieved nearly the same performance compared to
> > > without using Dropout. The results are consistent with the work of Dwivedi et al. (2020), wherein
> > > the rate for Dropout is set to in the base models, including GCN, GAT and GatedGCN.
> > >
> > > Additionally, the idea of Dropout is similar to the idea of our mutation method, which randomly
> > > replace a value with that sampled from a Gaussian distribution. The use of our mutation method
> > > improves the performance on the CIFAR10 dataset .
> > >
> > > Regarding analysis of overmoothing using metrics, we are still working on this. Given the discussion
> > > period is to be due shortly, we will be continue working on it after the discussion period.

---

> > > > ### Comment · Reviewer_uj5y · 2024-12-02
> > > > **responses**
> > > >
> > > > Hi,
> > > >
> > > > Thank you for your work; it is excellent. I look forward to seeing your evaluations on oversmoothing.

---

### Official Review · Reviewer_i1Dq · 2024-11-01

**Soundness:** 1
**Presentation:** 2
**Contribution:** 1
**Rating:** 3
**Confidence:** 4

**Summary:**

The paper deals with message passing on graph structures. The authors propose a method to tackle oversmoothing issue in the graph neural networks. They first posit that the main problem with over-smoothing is the lack of diversity in the embeddings, and propose to use techniques that can bias the embedding generation process to maintain diversity during message iterations. They propose an algorithm which mimics some operations from evolutionary theory i.e. crossover and mutation. Importantly, they introduce two crossover operations, i.e., generation crossover and sibling crossover, and a mutation operation, interleaved with the current message passing paradigm. Experiments on various datasets show that the method improves prediction capacity of GNNs.

**Strengths:**

1. The paper addresses an important problem of oversmoothing seen in GNNs..
1. The proposed crossover and mutation operations are interesting. However, the need and analysis of their effectiveness needs more detailed study.
1. The paper is easy to follow.

**Weaknesses:**

1. For evolutionary theory to work, it relies on randomness and scale i.e. over large number of iterations, small changes produce variety and adaptable traits.  How many iterations can GNN be applied for a reasonable analogy to the theoretical foundations of evolutionary theory is unclear.
1. The authors say - This makes it interpretable and easy for understanding. It is not clear how does this make it more interpretable, when it is making more random and introducing distinctness, even in cases when there is no need based on features or structure.
1. After Crossover operation and mutation, evolutionary pressures which select the fit traits and the rest are discarded i.e. fail to reproduce enough. However, I do not see how these selection is happening in the GNN framework. If there is no selection, then it means all crossover and mutations are good, which is contrary to evolutionary theory.
1. Typos: Line 125, 134, 208,

**Questions:**

Please see weaknesses.

---

> ### Author Response · Authors · 2024-11-28
> **Response to Reviewer i1Dq**
>
> **1. For evolutionary theory to work, it relies on randomness and scale i.e. over large number of iterations, small changes produce variety and adaptable traits. How many iterations can GNN be applied for a reasonable analogy to the theoretical foundations of evolutionary theory is unclear.**
>
> Response: Thank for the time reviewing our paper. We would like to clarify that this
> work proposes operations by drawing inspiration from genetic algorithms for addressing the over-smoothing problem, a common issue in most graph neural networks.
> The key operations, i.e, crossover and mutation, can be integrated into current graph neural networks, and the model can be optimized using backkpropagation.
> The number of iterations is the same as that used for optimizing the graph neural network model with backpropagation.
>
> **2. The authors say - This makes it interpretable and easy for understanding. It is not clear how does this make it more interpretable, when it is making more random and introducing distinctness, even in cases when there is no need based on features or structure.**
>
> Response: Thank you for the comment. We would like to clarify that this paper proposes operations by drawing inspiration from genetic algorithms for addressing the over-smoothing problem in graph neural network. Our framework has good interpretablilty as it views
> the layerwisely node embedding process as analogous to the genetic evolutionary process.
> The introduced operations,i.e, crossover and mutation, helps prevent
> the generated embeddings from becoming too similar to one another when more graph network
> layers are used.
>
> **3. After Crossover operation and mutation, evolutionary pressures which select the fit traits and the rest are discarded i.e. fail to reproduce enough. However, I do not see how these selection is happening in the GNN framework. If there is no selection, then it means all crossover and mutations are good, which is contrary to evolutionary theory.**
>
> Response: We would like to clarify that the use of the two operations, i.e., crossover and mutation, inspired by genetic algorithms prevents the generated embeddings from becoming too similar to one another when more graph network
> layers are used. The proposed framework helps to alleviate over-smoothing, a common issue in
> most current graph neural network models.
> Our framework does not use the selection mechanism.
> Our framework is not simply a combination of  genetic algorithms with graph neural networks.
>
> **4. Typos: Line 125, 134, 208,**
>
> Response: Thank you, the typose have been fixed.

---

### Official Review · Reviewer_6NCT · 2024-11-01

**Soundness:** 3
**Presentation:** 3
**Contribution:** 2
**Rating:** 3
**Confidence:** 5

**Summary:**

This paper proposes a new approach, called Genetic-Evolutionary Graph Neural Networks, to enhance graph representation learning by tackling the well-known over-smoothing issue in message-passing graph neural networks (MP-GNNs). The method introduces three key operations — cross-generation crossover, sibling crossover, and mutation — within a genetic evolution-inspired framework. These operations are designed to sustain diversity in node embeddings, providing interpretability and adaptability to existing MP-GNNs. The paper validates the model’s performance across six benchmark datasets, achieving ideal results on graph-related tasks.

**Strengths:**

Originality: This application of genetic-inspired methods to maintain embedding diversity is a creative approach.

Quality: The proposed framework is validated through experiments on multiple benchmark datasets, and the results indicate an improvement over the baseline models.

Clarity: The paper clearly outlines its methodology, with illustrative examples for the proposed genetic operations.

Significance: By offering a generalizable approach to improving MP-GNNs without adding complex parameters, this work has potential relevance to broader applications in graph learning.

**Weaknesses:**

1. I recommend that the authors include a review of existing literature in the introduction to highlight the contribution of this study and its distinctions from previous work.

2. The over-smoothing problem can be mitigated through graph structure learning, residual connections, and similar techniques. I suggest that the authors add baseline experiments with these approaches.

3. The Related Work section have not evolutionary-based GNNs, but there has been considerable research in this area, like literatures [1,2]. I recommend including a discussion of these relevant studies.

4. I suggest that the authors use a more general pseudocode format (e.g., a higher-level process description) instead of framework-specific syntax to better adhere to pseudocode standards—focusing on the logical flow rather than the framework implementation. This will enhance the paper’s accessibility and allow readers without a specific framework background to more easily understand the algorithm’s principles.

5. To improve the transparency and reproducibility of this research, I recommend that the authors consider open-sourcing their code to promote academic exchange and development in this field.

6. The Crossover and Mutation operations in this paper are not sufficiently innovative.

[1] Shi, Min, et al. "Genetic-gnn: Evolutionary architecture search for graph neural networks." Knowledge-based systems 247 (2022): 108752.
[2] Liu, Zhaowei, et al. "EGNN: Graph structure learning based on evolutionary computation helps more in graph neural networks." Applied Soft Computing 135 (2023): 110040.

**Questions:**

Please refer to the comments provided in the "Weaknesses" section for further details.

---

> ### Author Response · Authors · 2024-11-28
> **Response to Reviewer 6NCT**
>
> **1. I recommend that the authors include a review of existing literature in the introduction to highlight the contribution of this study and its distinctions from previous work.**
>
> *Response:* Thank you for the comment. We have added a brief comparison of our work with  previous work in the introduction section as follows. "Unlike previous methods, such as residual connections (He et al., 2016), SSFG (Zhang et al., 2022)
> and PariNorm (Zhao \& Akoglu, 2020), this work proposes operations by drawing inspiration from genetic algorithms for addressing the over-smoothing problem in graph neural network.
> Our framework has good interpretablilty as it views the layerwisely node embedding process as analogous to the genetic evolutionary process."
>
> **2. The over-smoothing problem can be mitigated through graph structure learning, residual connections, and similar techniques. I suggest that the authors add baseline experiments with these approaches.**
>
> *Response:*  Thanks a lot for the suggestion.
> The baseline models of GPS/Finetuned GPS, GCN, SSFG, GatedGCN already used the techniques, such as residual connections, batch normalizations and structural encoding. Otherwise the model performance would be more significantly reduced.
> We have further compared our method with the basic GCN in which residual connections and batch normalizations (BN) are not used on MNIST and CIFAR10.
> The results have been added in Table 5 in our revised manuscript.
> It can be seen from Table 5 that the model performance drops without using these techniques and that the joint use of our method with residual connections and batch normalizations yields the best task performance.
>
>
> **3. The Related Work section have not evolutionary-based GNNs, but there has been considerable research in this area, like literature [1,2]. I recommend including a discussion of these relevant studies.**
>
> *Response:* We have included the relevant studies in the Related Work section. The two studies both use the evolutionary method to search and optimize graph neural network models and hyperparameters.
> Unlike these studies, we use the genetic-evolutionary strategy to maintain embedding diversity  for addressing the oversmoothing problem in the node embedding generation process. It is a general method that can be applied in different graph nerual network models.
>
> **4. I suggest that the authors use a more general pseudocode format (e.g., a higher-level process description) instead of framework-specific syntax to better adhere to pseudocode standards—focusing on the logical flow rather than the framework implementation. This will enhance the paper’s accessibility and allow readers without a specific framework background to more easily understand the algorithm’s principles.**
>
> *Response:* Thank you. We have added a more general pseudocode for the operations of crossover and mutation in the appendix section (see Algorithm 3 and Algorithm 4).
>
>
> **5. To improve the transparency and reproducibility of this research, I recommend that the authors consider open-sourcing their code to promote academic exchange and development in this field.**
>
> *Response:* We have added the code of the crossover and mutation functions in Pytorch to the appendix section. The full project code will be made publicly available if this work could be accepted for publication.
>
>
> **6. The Crossover and Mutation operations in this paper are not sufficiently innovative.**
>
> *Response:* Thank you for the comment. We would like to clarify that, to the best of our knowledge, this is the first work of integrating the genetic-evolutionary strategy into graph neural network for  node embedding generation. The use of the two operations inspired by genetic algorithms prevents the generated embeddings from becoming too similar to one another when more graph network layers are used.
> The proposed framework helps to alleviate over-smoothing, a common issue in most current graph neural network models.
> The results show that our method achieves performance improvements on strong baseline models on different datasets.

---

> > ### Comment · Reviewer_6NCT · 2024-12-01
> >
> > Thank you to the authors for their responses. However, I still have the following concerns:
> >
> > 1. This paper does not fully clarify the concept of Genetic-Evolutionary. For example, to my understanding, the process of evolution/genetic should also include Evaluation (to evaluate the quality or potential of each offspring based on certain strategies) and Selection (to choose offspring based on their fitness or performance). Without these components, the process seems incomplete.
> >
> > 2. The Crossover operation is a random selection process rather than an iterative optimization process. Therefore, is it reasonable to interpret the proposed model’s optimal performance as a result of coincidental randomness? Additionally, I suggest the authors include a clearly defined objective function.
> >
> > 3. It appears that the authors are avoiding discussing Evolutionary-based GNNs. There are already numerous similar GNN papers published in this area. Why did the authors not cite any of them?
> > 4. To my knowledge, one natural advantage of introducing Evolutionary Algorithms into GNNs is to provide diversity for node embeddings. However, the contribution of this work seems to merely reintroduce Evolutionary/Genetic approaches into GNNs to address overfitting caused by multi-layer architectures. This method lacks originality and depth, and I do not believe it meets the standards of a top-tier conference like ICLR. The authors might consider submitting this work to a lower-tier conference or journal.
> >
> > 5. I recommend the authors conduct a thorough review of the latest literature. It is evident that this is not the first work integrating the Genetic-Evolutionary strategy into Graph Neural Networks for node embedding generation.

---

> ### Author Response · Authors · 2024-12-02
> **Response to Reviewer 6NCT (Part 1)**
>
> **1. This paper does not fully clarify the concept of Genetic-Evolutionary. For example, to my understanding, the process of evolution/genetic should also include Evaluation (to evaluate the quality or potential of each offspring based on certain strategies) and Selection (to choose offspring based on their fitness or performance). Without these components, the process seems incomplete.**
>
> *Response:* Thank you very much for the comments. We are sorry for the confusion. We would like to clarify that this paper is not simply a combination of genetic/evolutionary algorithms and graph neural networks. This paper draws inspiration from genetic algorithms and views the node embedding generation process as a genetic-evolutionary process, wherein each of the graph neural network layers produces a new generation of embeddings.
>
> To prevent the embeddings become very similar to one another, we introduce operations inspired by crossover and mutation to maintain the embedding diversity.
> The crossover and mutation operation introduced in this paper can be integrated into current graph neural network models (the Pytorch implementations of the key operations are included in the appendix section).
> Otherwise if node embeddings become very identical to each other, the original input node attributes or features become lost, and the model performance can be significantly reduced.
>
> To summarize, this paper proposes operations, i.e., crossover and mutation, by drawing inspiration from genetic algorithms, and it is not a simply use of genetic algorithms for node embedding generation.
> The proposed operations are used to maintain or increase the diversity of generated embeddings.
> We don's use selection or evaluation of selected embeddings.
> The selection and evaluation strategies are commonly used for search purposes.
> These strategies are not used in our embedding generation problem.
>
>
> **2. The Crossover operation is a random selection process rather than an iterative optimization process. Therefore, is it reasonable to interpret the proposed model’s optimal performance as a result of coincidental randomness? Additionally, I suggest the authors include a clearly defined objective function.**
>
> *Response:* Thank you for the comment. The Crossover operation is a random selection process, and the Mutation operation also increases randomness. The two operations help prevent node embeddings from becoming similar to one another.
> However, it is not simply a results of coincidental randomness. We evaluated the method of Dropout on MNIST and CIFAR10 and found that the use of Dropout achieved nearly the same performance compared to without using Dropout.
> The results are consistent with the work of Dwivedi et al. (2020), wherein the rate for Dropout is set to in the base models, including GCN, GAT and GatedGCN.
> Also, from Figure 5 in the appendix section, it can be seen that the model performance improves as the number of graph neural network layers increases from 4 to 12 with out method. Whereas without using our method, the performance drops as the layer number increase, a common phenomenon as a result of over-smoothing .
>
>
> **3. It appears that the authors are avoiding discussing Evolutionary-based GNNs. There are already numerous similar GNN papers published in this area. Why did the authors not cite any of them?**
>
> *Response:* We apologize for  uploading not the latest revised version. We thought we uploaded the version that has discussions about the two related studies you kindly suggested. We have only now found out that we didn't uploaded the latest version. If we could be given the chance, we will upload the version with discussing about evolutionary studies inducing the two you kindly recommended  in graph neural networks.
>
> The work of Shi et al. (2022) uses the evolutionary approach for search graph neural network architectures as well as hyperparameters.
> The work of Liu et al. (2023) adopts the evolutionary approach for optimizing graph structures, enabling the graph neural network model can better defend againast adversarial attacks.
>
> [1] Shi, Min, et al. "Genetic-gnn: Evolutionary architecture search for graph neural networks." Knowledge-based systems 247 (2022): 108752.
>
> [2] Liu, Zhaowei, et al. "EGNN: Graph structure learning based on evolutionary computation helps more in graph neural networks." Applied Soft Computing 135 (2023): 110040.

---

> ### Author Response · Authors · 2024-12-02
> **Response to Reviewer 6NCT (Part 2)**
>
> **4. To my knowledge, one natural advantage of introducing Evolutionary Algorithms into GNNs is to provide diversity for node embeddings. However, the contribution of this work seems to merely reintroduce Evolutionary/Genetic approaches into GNNs to address overfitting caused by multi-layer architectures. This method lacks originality and depth, and I do not believe it meets the standards of a top-tier conference like ICLR. The authors might consider submitting this work to a lower-tier conference or journal.**
>
> *Response:* Thank you for the comment. However, we don't agree. Firstly, our method does not simply address the overfitting problem. In the right of Figure 4, it can be seen that our method improves both the training performance and test performance. It improves the overall graph representation learning performance, which is not only by addressing overfitting.
> As mentioned in our  responses to your question 2, we evaluated the method of Dropout on MNIST and CIFAR10 and found that the use
> of Dropout achieved nearly the same performance compared to without using Dropout. The use of the Dropout regularization method does not improve the model performance. Wheras our method can effectively improve the performance on different datasets. It can address overfitting and underfitting for different tasks (see Figure 4 and Figure 5).
>
> **5. I recommend the authors conduct a thorough review of the latest literature. It is evident that this is not the first work integrating the Genetic-Evolutionary strategy into Graph Neural Networks for node embedding generation.**
>
> *Response:* Thank you for the comment, and we apologize again for uploading not latest version.
> The work of Shi et al. (2022) uses the evolutionary approach for search graph neural network architectures as well as hyperparameters. The work of Liu et al. (2023) adopts the evolutionary approach for optimizing graph structures, enabling the graph neural network model can better defend againast adversarial attacks.
> In object detection, evolutionary approach was also used for optimizing hyperparameters in latest YOLO models
> (Redmon, 2016). Unlike these studies, which primary use genetic evolution as an approach to search for the graph structures, or graph neural network architectures or hyperparameters.
> This paper draws inspiration from genetic algorithms and views the node
> embedding generation process as a genetic-evolutionary process, wherein each of the graph neural
> network layers produces a new generation of embeddings.
> We introduce key operations, i.e., crossover and mutation, inspired by evolutionary algorithms to maintain the embedding diversity, which prevents embedding from becoming similar to one another, a common issue in graph neural networks.
>
>
> [1] Shi, Min, et al. "Genetic-gnn: Evolutionary architecture search for graph neural networks." Knowledge-based systems 247 (2022): 108752.
>
> [2] Liu, Zhaowei, et al. "EGNN: Graph structure learning based on evolutionary computation helps more in graph neural networks." Applied Soft Computing 135 (2023): 110040.
>
> [3] Redmon, Joseph, et al. "You only look once: Unified, real-time object detection." In Proceedings of the IEEE
> conference on computer vision and pattern recognition, 2016

---

> > ### Comment · Reviewer_6NCT · 2024-12-03
> >
> > Thank you to the authors for their response.

---

### Official Review · Reviewer_RrGT · 2024-11-02

**Soundness:** 2
**Presentation:** 2
**Contribution:** 3
**Rating:** 6
**Confidence:** 4

**Summary:**

The paper proposes a novel framework for enhancing graph neural network (GNN) models by integrating genetic algorithm concepts such as crossover and mutation into the training process. This approach aims to combat the pervasive over-smoothing problem in conventional GNNs by maintaining diversity in node embeddings, thereby allowing the model to capture more complex graph structures and relationships. The results from experiments on several benchmark datasets indicate significant performance improvements over existing methods, making a compelling case for the use of evolutionary strategies in graph representation learning.

**Strengths:**

originality: medium
quality: good
clarity: good
significance: medium

**Weaknesses:**

see below

**Questions:**

1. "it is important to perserve the diversity of generated embeddings throughout their layerwisely generation process." It is true that a GNN with good performance needs some extent of diversity in the node embedding. However, is it also true that diverse embedding will definitely lead to good performance? In other words, is diverse embedding a sufficient condition for good GNN performance?

2. Equation (2) and (3) look like random pooling operations. Is there any relation?

3. How many layers does the model have in your experiments?

4. How does your model performance changes as the layer goes to deep? Need ablation study for different operations in your model.

5. How does your model work on node classification problem, especially on heterophilic graphs[1]?




[1] When Do Graph Neural Networks Help with Node Classification? Investigating the Homophily Principle on Node Distinguishability. Advances in Neural Information Processing Systems. 2024 Feb 13;36.

---

> ### Author Response · Authors · 2024-11-28
> **Response to Reviewer RrGT**
>
> **1. "It is important to preserve the diversity of generated embeddings throughout their layerwisely
> generation process." It is true that a GNN with good performance needs some extent of diversity
> in the node embedding. However, is it also true that diverse embedding will definitely lead to
> good performance? In other words, is diverse embedding a sufficient condition for good GNN
> performance?**
>
> *Response:* Thank you for the comment. The issue of over-smoothing is a common issue in graph neural networks.
> The embeddings of nodes tend to become very similar to one another after repeatedly apply graph convolutions due to the over-smoothing problem. The consequence is that the original input node attributes or features become lost, and the model performance can be significantly reduced. In this paper, by "preserving the diversity of generated embeddings", the model basically prevents the node embeddings becoming too similar to one another, which prevents the issue of the input node attributes or features being washed out.
>
> From another perspective, we view the layerwisely embedding generation process as an evolutionary process.
> The methods of crossover and mutation are introduced to generate diversed embeddings (or populations) than very homogeneous ones. Homogeneous populations are regarded as a major issue for premature converging to suboptimal solutions (Whitley, 2001) in genetic algorithms. By preserving the diversity of embeddings, the model can be prevented from premature converging to local minima.
>
> **2. Equation (2) and (3) look like random pooling operations. Is there any relation?**
>
> *Response:* In the model training stage, the feature at each position is randomly selected from one of the input vectors (in Equations (2) and (3)). This operation is essentially the same as the randomly pooling operation.
> The difference is that, in the model inference stage, we use a deterministic output, which is computed based on the sampling factor (see Algorithm 1).
>
> **3. How many layers does the model have in your experiments?**
>
> *Response:* Finetuned GPS model.
> The number of layers is 6 and 8 on MNIST and CIFAR10 respectively, and the number of layers is set to 8 on PascalVOC-SP and COCO-SP. In Table 3, our results on Finetuned GCN also uses the same of number of layers (6 layers) as the base Finetuned GCN model.
>
> **4. How does your model performance changes as the layer goes to deep? Need ablation study for different operations in your model.**
>
> *Response:* We have further conducted experiments to analysis the performance of our method on
> the base Finetuned GPS model with different layers on CIFAR10 and PascalVOC-SP. The results
> have been added in Figure 4 in the revised version. We have also analyzed the the performance of
> our method on the base Finetuned GPS model with different layers on CIFAR10. The results have
> been added in Figure 5 in the appendix section. It can be seen from Figure 4 and Figure 5 that the
> use of our method improves the model generalization performance of the base models.
>
> **5. How does your model work on node classification problem, especially on heterophilic graphs[1]?**
>
> *Response:* We would like to clarify that  Table 2 already showed the results on  node classification on PascalVOC-SP and COCO-SP.
> The two datasets contain superpixel graphs, and the task is  to predict if a node corresponds to a region of an image which belongs to a particular class.
> We used the same model hyper-parameters as the base Fine-tuned GPS model and achieved improved performance compared with the base model on the node classification task on the two datasets.
>
> Additionally, we have discussed the work of  Luan et al. (2024) about the impact of graph homophily on learning using graph neural networks.

---

> > ### Comment · Reviewer_RrGT · 2024-12-01
> > **Thanks for the reply**
> >
> > Thanks for the reply from the authors. Most of my concerns have been addressed. Although there is still some room for improvement, I think the authors have proposed an interesting method that is simple and effective. I will increase my rating from 5 to 6. Good luck.

---

> > > ### Author Response · Authors · 2024-12-02
> > >
> > > Thank you very much. We will be futher improving our work based on your valuable comments and suggestions.

---

### Meta-Review · Area_Chair_FP58 · 2024-12-17

**Metareview:**

This paper presents a method for addressing  the issue of over-smoothing in message-passing GNN. The paper argues that this problem comes from a lack of diversity in the generated embeddings. The paper proposes to use a novel framework inspired by genetic algorithms called genetic-evolutionary graph neural networks and using the principles of crossover and mutations to promote diversity in the embeddings

Strengths:
- interesting solution for the issue of over-smoothing in GNN
- clear paper,
- novelty of the propositions
- improvements of a baseline.

Weaknesses:
- some elements need more discussion/justifications, the need of diversity should be better justified,
- review of existing literature can be improved,
- additional baselines should be considered,
- parts of the experimental evaluation could be improved (like other analysis of oversmoothing using metrics such as Mean Absolute Error (MAE) or Dirichlet Energy).

Authors have provided different answers during the rebuttal, including additional results but some elements were still missing.

Reviewers have mentioned that the paper provides an interesting and effective method. However, the literature review can still be enlarged and improved. The experimental evaluation on over-smoothing can be improved which is key for this paper since it is the core of the contribution.
If the paper has important merits, there was no strong consensus for acceptance in its current version.

I propose then rejection.

**Additional Comments On Reviewer Discussion:**

Authors have provided different answers during the rebuttal, including additional results but not on all the points raised by reviewers.
Reviewers RrGT were satisfied by the answers and improved his score.
Reviewer 6NCT acknowledged authors' answers without improving his score.
Reviewer uj5y was satisfied of a part of the answers but required other experimental evaluation on over-smoothing.

Overall, while the paper has merits and authors did a strong effort for the rebuttal, there was no consensus for proposing acceptance.

---

### Decision · Program_Chairs · 2025-01-22

Reject